# Liver-specific ATP-citrate lyase inhibition by bempedoic acid decreases LDL-C and attenuates atherosclerosis

Stephen L. Pinkosky[1,2], Roger S. Newton[1], Emily A. Day[2], Rebecca J. Ford[2], Sarka Lhotak[3], Richard C. Austin[3], Carolyn M. Birch[1], Brennan K. Smith[2], Sergey Filippov[1], Pieter H.E. Groot[1], Gregory R. Steinberg[2,4,*] & Narendra D. Lalwani[1,*]

Despite widespread use of statins to reduce low-density lipoprotein cholesterol (LDL-C) and associated atherosclerotic cardiovascular risk, many patients do not achieve sufficient LDL-C lowering due to muscle-related side effects, indicating novel treatment strategies are required. Bempedoic acid (ETC-1002) is a small molecule intended to lower LDL-C in hypercholesterolemic patients, and has been previously shown to modulate both ATP-citrate lyase (ACL) and AMP-activated protein kinase (AMPK) activity in rodents. However, its mechanism for LDL-C lowering, efficacy in models of atherosclerosis and relevance in humans are unknown. Here we show that ETC-1002 is a prodrug that requires activation by very long-chain acyl-CoA synthetase-1 (ACSVL1) to modulate both targets, and that inhibition of ACL leads to LDL receptor upregulation, decreased LDL-C and attenuation of atherosclerosis, independently of AMPK. Furthermore, we demonstrate that the absence of ACSVL1 in skeletal muscle provides a mechanistic basis for ETC-1002 to potentially avoid the myotoxicity associated with statin therapy.

[1] Esperion Therapeutics, Inc., 3891 Ranchero Drive, Suite 150, Ann Arbor, Michigan 48108, USA. [2] Division of Endocrinology and Metabolism, Department of Medicine, McMaster University, 1280 Main Street West, Hamilton, Ontario, Canada L8S 4K1. [3] Department of Medicine, McMaster University, St Joseph's Healthcare Hamilton, 50 Charlton Avenue East, Hamilton, Ontario, Canada L8N 4A6. [4] Department of Biochemistry and Biomedical Sciences, McMaster University, 1280 Main Street West, Hamilton, Ontario, Canada L8S 4K1. * These authors contributed equally to this work. Correspondence and requests for materials should be addressed to S.L.P. (email: spinkosky@esperion.com) or to G.R.S. (email: gsteinberg@mcmaster.ca).

An elevated level of plasma low-density lipoprotein cholesterol (LDL-C) is a significant risk factor for atherosclerotic cardiovascular disease (ASCVD), the leading cause of death and disability in the western world[1,2]. Statins are the standard of care for controlling elevated LDL-C and are proven to reduce cardiovascular risk and prevent progression of coronary heart disease[3–5]. Statins lower LDL-C by inhibiting hepatic 3-hydroxy-3-methyl-glutaryl-CoA (HMG-CoA) reductase, the rate-limiting enzyme in the cholesterol biosynthesis pathway[6]. This inhibition leads to reduced hepatic cholesterol levels, which triggers the upregulation of LDL receptors (LDLR) resulting in increased LDL particle clearance from the blood[7–9]. Despite the proven benefits of statins, many patients remain at risk for ASCVD due to their inability to tolerate the statin dose required to achieve recommended LDL-C goals. Myalgia (muscle pain, cramping and/or weakness) constitutes the most common adverse effect associated with statin treatment and often results in dose limitations, poor compliance or even discontinuation[10,11]. An estimated 2–7 million patients in the United States have stopped statin treatment due to muscle complaints despite being at risk for cardiovascular disease[12,13]. Although the underlying pathophysiology of statin-induced myalgia is not completely understood, significant evidence supports that it is linked to HMG-CoA reductase inhibition in skeletal muscle resulting in reduced production of one or more biological intermediates important to maintain normal muscle cell function[14–19]. This insight has attracted interest in identifying novel treatment strategies capable of complementing the LDL-C-lowering effects of statins without blocking the biosynthesis of key products required for normal skeletal muscle function.

ATP-citrate lyase (ACL) is a cytosolic enzyme upstream of HMG-CoA reductase in the lipid biosynthesis pathway that catalyses the cleavage of mitochondrial-derived citrate into oxaloacetate and acetyl-CoA, the latter serving as common substrate for de novo cholesterol and fatty acid synthesis. Although ACL is not rate limiting, its strategic position at the intersection of lipid and carbohydrate metabolism, and its potential to regulate lipoprotein metabolism, attracted early interest as a drug target to treat dyslipidemia[20–23]. Early discovery strategies that focused on synthesizing citrate and citryl-CoA analogues yielded compounds that were highly potent in cell-free systems; however, they did not advance to clinical development largely due to poor cell permeability and insufficient bioavailability[24,25].

ETC-1002 (bempedoic acid; 8 hydroxy-2,2,14,14 tetramethyl-pentadecanedioic acid) is a first-in-class, oral, small-molecule cholesterol synthesis inhibitor that appears to circumvent these issues by exploiting the activity of an unknown endogenous acyl-CoA synthetase (ACS) to mediate its intracellular CoA activation to the active ACL inhibitor, ETC-1002-CoA (ref. 26). Consistent with inhibition of cholesterol synthesis, ETC-1002 significantly lowers elevated levels of LDL-C in hypercholesterolemic patients by 30% as monotherapy, up to an additional 24% when added on to stable statin therapy and up to 50% when combined with ezetimibe, suggestive of a distinct mechanism for LDL lowering[27–30]. We have previously shown that the hypolipidemic effects of ETC-1002 are consistent with the inhibition of hepatic ACL by the CoA thioester form of ETC-1002 (ETC-1002-CoA), which results in the suppression of metabolic intermediates downstream of ACL and a reduction in the rates of de novo cholesterol and fatty acid synthesis. However, we have also demonstrated that ETC-1002 treatment increased AMP-activated protein kinase (AMPK)[26], a metabolic sensor capable of catalysing regulatory phosphorylation of numerous substrates that affect inflammatory signalling and lipid metabolism[31–39]. This activation of AMPK in the liver of mice was not associated with changes in cellular energy charge suggesting that ETC-1002 may directly increase AMPK activity via an allosteric mechanism. Despite the importance of these studies, the mechanism by which ETC-1002 lowers LDL-C and modulates ACL and AMPK activity, and the potential importance of these pathways remain undefined. Furthermore, it is currently unknown whether LDL-C lowering by ETC-1002 is sufficient to reduce the progression of atherosclerosis.

In the present study, we establish ETC-1002 as a prodrug, which requires the activity of very-long-chain acyl-CoA synthease-1 (ACSVL1) for conversion to an active modulator of ACL and AMPK activity. Using genetic and pharmacologic methods to suppress ACL in vitro, and by testing the effects of ETC-1002 in a novel APOE/AMPK β1 double-knockout (DKO) mouse model, we establish ACL inhibition as the primary mechanism leading to reduced LDL-C and atherosclerosis. Furthermore, we provide a mechanistic basis for the differentiation of ETC-1002 from the myotoxic effects associated with statins by demonstrating the absence of ACSVL1 expression in skeletal muscle from mice and humans, and showing that ETC-1002 does not suppress the cholesterol biosynthesis pathway in this tissue nor promote the associated myotoxicity.

## Results

**ETC-1002-CoA directly modulates ACL and AMPKβ1 activity.** To investigate the mechanism by which ETC-1002 inhibits ACL and increases AMPK activity, we tested whether ETC-1002 or ETC-1002-CoA directly modulated recombinant human ACL and AMPK$\alpha_1\beta_1\gamma_1$ heterotrimeric complexes in cell-free systems. We first conducted kinetic analyses of ETC-1002 and ETC-1002-CoA against multiple ACL substrates and coenzymes to better understand the molecular mechanism of inhibition. These studies showed that ETC-1002-CoA exhibited competitive inhibition kinetics with respect to CoA (Fig. 1a) ($K_i = 2\,\mu M$), while non-competitive inhibition was observed for citrate (Fig. 1b) and ATP (Fig. 1c), suggesting that ETC-1002-CoA competes for CoA binding. Importantly, ETC-1002 free acid was confirmed to be inactive against recombinant human ACL (Fig. 1d).

We next investigated how ETC-1002 activated AMPK. Using purified enzyme preparations of AMPK$\alpha_1\beta_1\gamma_1$, we found that ETC-1002-CoA dose-dependently activated the kinase, while ETC-1002 free acid had no effect (Fig. 1e). As anticipated, A-769662 (a direct β1-specific activator of AMPK[40]) and AMP increased enzyme activity (Fig. 1e). We then completed the same assays in AMPK α1β2γ1 complexes and found that while AMP continued to activate the complex, both ETC-1002-CoA and A769662 did not increase enzyme activity (Fig. 1f). These findings in cell-free enzyme assays (1) establish that ETC-1002-CoA is a competitive ACL inhibitor with respect to free CoA (2) demonstrate that ETC-1002-CoA is the active form that directly interacts with AMPK, and not the free acid as previously proposed[26] and (3) show that ETC-1002 can only inhibit ACL and activate AMPK β1 in tissues that are capable of catalysing the CoA activation of ETC-1002 depicted in (Fig. 1g).

**Identification of the ETC-1002 synthetase.** Given the requirement for CoA activation of ETC-1002 to inhibit ACL and activate AMPK, we performed a series of studies aimed to identify the specific ACS isoform that catalyses this reaction to better define the tissue specificity for ETC-1002 activity. Previous findings demonstrated the presence of ETC-1002-CoA in rodent liver[26]; therefore, we confirmed ETC-1002-CoA formation in human liver and demonstrate that the enzyme is almost exclusively present in the microsome-enriched fraction, while negligible

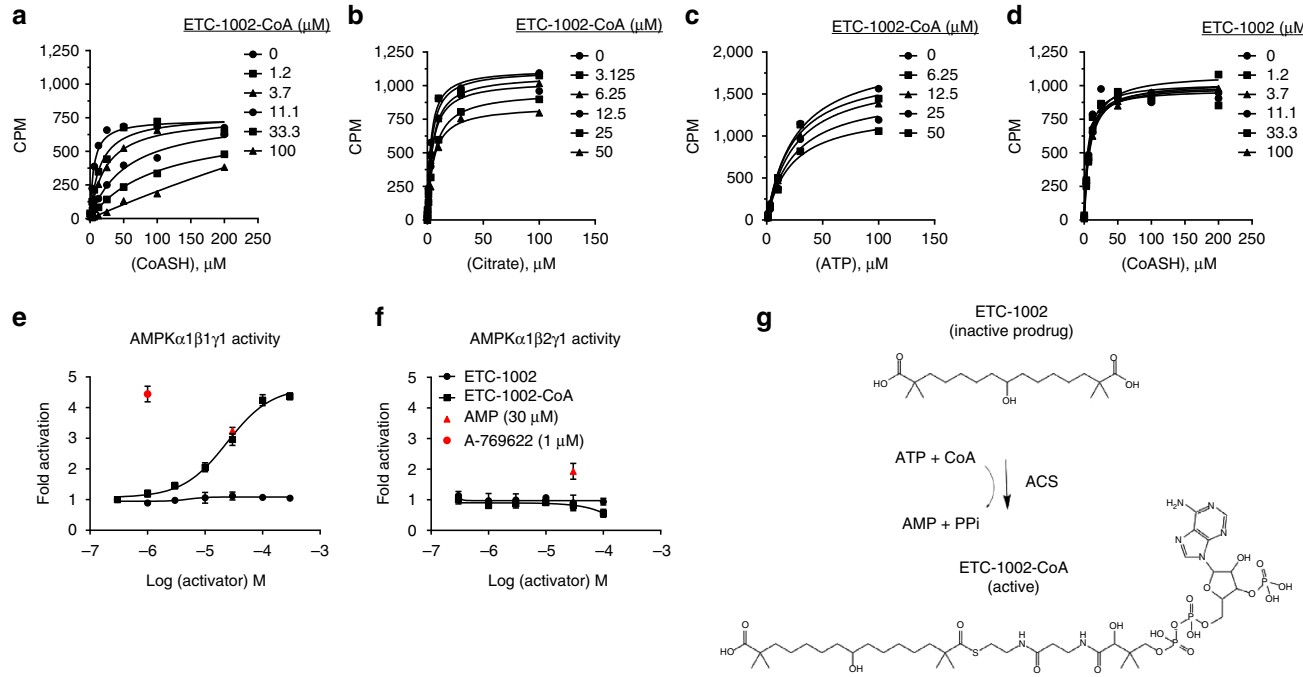

**Figure 1 | ETC-1002-CoA inhibits ACL and mediates β1-selective AMPK activation.** Recombinant human ACL was incubated in the presence of the indicated concentrations of ETC-1002-CoA (**a–c**) or ETC-1002 (**d**), and (**a**) coenzyme A (CoA), (**b**) citrate or (**c**) ATP. Conversion of [$^{14}$C]-citrate to [$^{14}$C]-acetyl-CoA was measured in counts per minute (CPM). Recombinant human (**e**) AMPKα1β1γ1 and (**f**) AMPKα1β2γ1 complexes were incubated in the presence of the indicated concentrations of ETC-1002, ETC-1002-CoA, AMP or A-769622. (**g**) Structure of ETC-1002 and the biochemical reaction that generates ETC-1002-CoA acyl-CoA synthetase (ACS). ETC-1002-CoA/CoASH enzyme ACL kinetic analyses are single measures, and representative results from $n = 3$ independent experiments shown; AMPK activity was determined by time-resolved fluorescence resonance energy transfer expressed as mean ± s.e.m. of triplicate measures. (**a**) Ki calculated by Michaelis–Menten kinetic analysis.

activity was detected in the mitochondrial- or cytosol-enriched fractions (Supplementary Fig. 1A). We then characterized ACS kinetics in human liver microsomes to establish optimum assay conditions (Supplementary Fig. 1B), and demonstrated consistent ETC-1002-CoA synthesis among individual human donors ($784 \pm 124$ s.e.m. pmol mg$^{-1}$ min$^{-1}$, $n = 8$).

Although the specific physiological roles of many ACS isoforms remain unknown, their respective substrate specificities, subcellular localization and tissue expression profiles have been extensively characterized[41,42]. To define the natural substrate profile of the ACS responsible for synthesizing ETC-1002-CoA, we measured [$^{14}$C]-ETC-1002 synthesis in the presence of a threefold molar excess of multiple unlabelled natural short-, medium-, long- and very-long-chain saturated fatty acids. These studies clearly show that fatty acids with C12 to C20 carbon chain lengths were most competitive for ETC-1002, with C22 fatty acid and larger also showing competition (Fig. 2a). When similar studies were carried out using aliphatic dicarboxylic acids as competitive substrates, near complete inhibition of ETC-1002-CoA synthesis was observed with C16 and C18 carbon lengths (Fig. 2b). As shown by Lineweaver–Burk plots, representative substrates for both C16 saturated mono- and dicarboxylic acids (that is, palmitic and hexadecanedioic acids) demonstrated competitive inhibition kinetics for ETC-1002-CoA synthesis (Supplementary Fig. 1C). Furthermore, the known ACSVL1 substrates, docosahexaenoic acid and eicosapentaenoic acid, also markedly competed for ETC-1002-CoA synthesis, while bile acid ligase activity, indicative of the liver-specific ACSVL6/FATP5 activity, was ruled out as cholate and chenodeoxycholate were shown to be inactive (Supplementary Fig. 1D). Members of the thiazolidinedione class of drugs, which have been shown to be specific ACSL4 inhibitors[43], also did not compete for

ETC-1002-CoA synthesis (Supplementary Fig. 1D). We have previously shown that triacsin C inhibited ETC-1002-CoA formation by ~40–50% in primary rat hepatocytes, and attenuated the inhibitory effects of ETC-1002 treatment on lipid synthesis to a similar degree[26]. Triacsin C is a potent inhibitor of ACSL1 and 4, and a weak inhibitor of ACSVL1 (refs 43–47). Therefore, we tested whether triacsin C directly inhibited ETC-1002-CoA synthesis in human liver microsomes. Consistent with the effects observed in hepatocytes, triacsin C (10 μM) inhibited ETC-1002-CoA formation by 40–50% (Supplementary Fig. 1D). When taken together, these competitive substrate and ACS inhibitor findings indicate a profile most consistent with the activities described for ACSVL1 (also known as FATP2; gene *Slc27a2*). This was further supported by showing that ACSVL1 protein levels correlated ($r^2 = 0.984$) to the rate of ETC-1002-CoA synthesis in various subcellular liver fractions (Supplementary Fig. 1E). To establish that ACSVL1 was the specific ACS isoform that catalyses the CoA activation of ETC-1002 we used small-interfering ribonucleic acid (siRNA)-mediated gene silencing in McArdle cells. ACSVL1 (*Slc27a2*) siRNA treatment reduced protein levels by more than 80%, which was associated with an 85% reduction in ETC-1002-CoA formation (Fig. 2c). Importantly, ACSVL1 gene silencing blocked ETC-1002 but not atorvastatin-dependent inhibition of *de novo* cholesterol synthesis, thus establishing the high specificity of ACSVL1 for the CoA activation of ETC-1002 (Fig. 2d). These studies demonstrate that the prodrug ETC-1002 is converted to ETC-1002-CoA by ACSVL1 and that this is required to suppress cholesterol synthesis in hepatocytes.

The established requirement for CoA activation suggests that ETC-1002 only modulates ACL and AMPK activities in cell types that express ACSVL1. Importantly, the tissue expression profile of

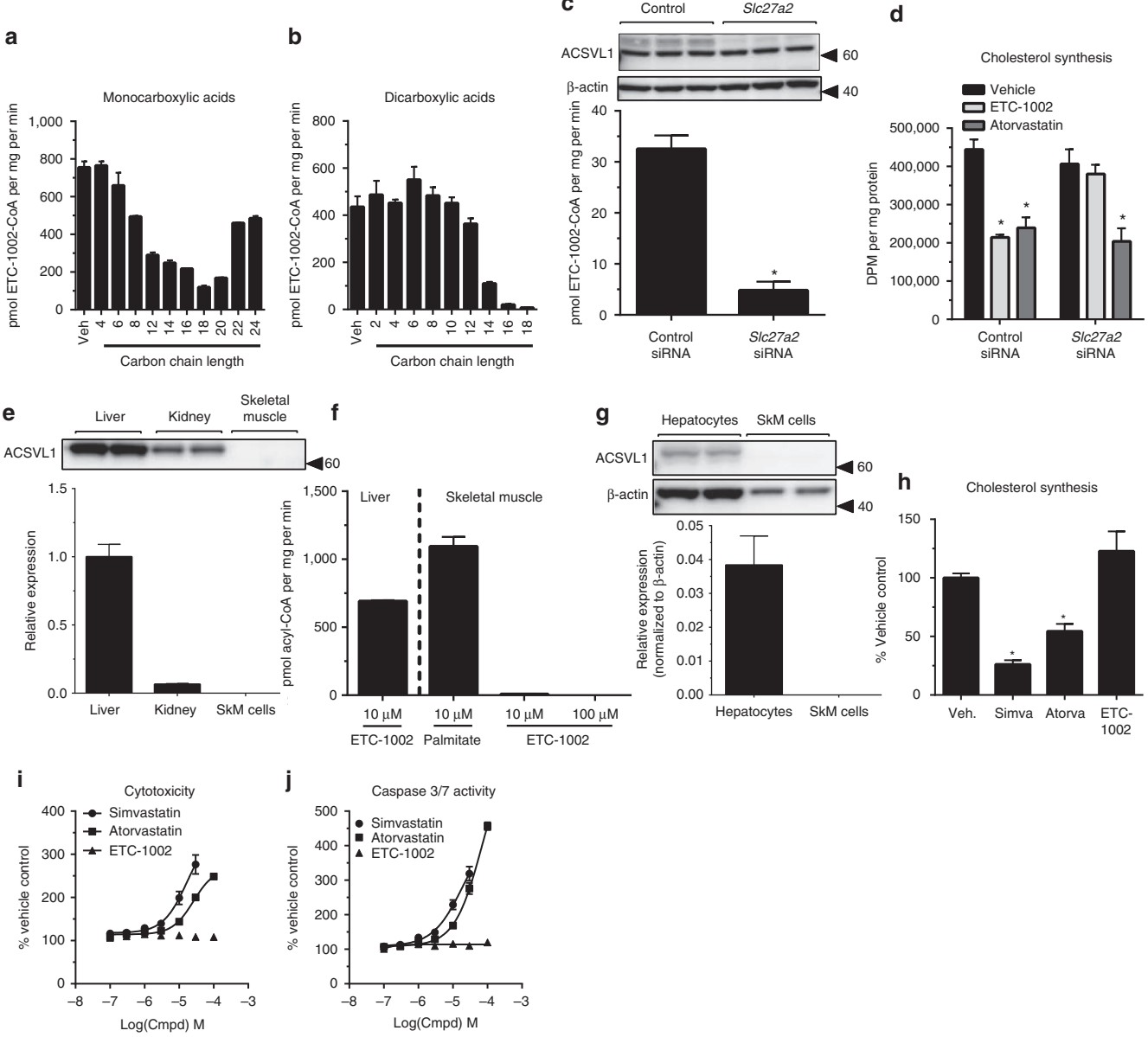

**Figure 2 | Differentiation of ETC-1002 from the myotoxic effects of statins.** [$^{14}$C]-ETC-1002 (10 μM) conversion to [$^{14}$C]-ETC-1002-CoA in human liver microsomes was determined in the presence of 30 μM unlabelled (**a**) monocarboxylic and (**b**) dicarboxylic saturated competitive fatty acid substrates. RH7777 cells were treated with negative control (Control) or *Slc27a2* (ACSVL1) siRNA; $n = 6$, for 48 h, and (**c**) ACSVL1 expression determined by western blot, and [$^{14}$C]-ETC-1002-CoA synthesis determined; $n = 3$. (**d**) Both control and *Slc27a2* siRNA-treated RH7777 cells were treated with vehicle, ETC-1002 (30 μM) or atorvastatin (0.5 μM), and *de novo* cholesterol synthesis measured over 4 h; $n = 6$ transfections. (**e**) Relative ACSVL1 expression in microsomes prepared from human liver, kidney and skeletal muscle (SkM). (**f**) ETC-1002-CoA synthetase activity in microsomes prepared from human liver and skeletal muscle; palmitate (10 μM) used as a positive control for skeletal muscle microsome viability. (**g**) ACSVL1 expression was measured in primary human hepatocytes and primary human skeletal muscle (SkM) cells by western blotting and expressed relative to β-actin. (**h**) Primary human myotubes were treated with vehicle (Veh.), simvastatin 10 μM (Simva), atorvastatin 10 μM (Atorva) or ETC-1002 100 μM in the presence of [$^{14}$C]-glucose for 12 h, and incorporation into non-saponifiable lipids determined; or treated with 0.1–100 μM simvastatin, atorvastatin or ETC-1002 for 48 h and (**i**) cytotoxicity was measure by GF-AFC/bis-AAF-R110 cleavage, and (**j**) Caspase 3,7 activity ( DEVD cleavage) determined. Data for myotube cytotoxicity assays are expressed as mean ± s.e.m., $n = 2$ performed in triplicate. Data for liver ($n = 50$ donors pooled) and kidney ($n = 50$ donors pooled) microsome preparations are representative of two independent experiments performed in duplicate, and expressed as mean ± s.d. Data for human skeletal muscle microsomes ($n = 4$ pooled donors) are the mean ± s.e.m. of two independent experiments performed in triplicate. Multiple comparisons were made using an one-way ANOVA, Bonferroni's multiple comparisons test; *$P < 0.05$.

ACSVL1 has been independently studied in mice and shown to be restricted to liver and kidney while absent in other tissues, including skeletal muscle[48]. Consistent with these previous observations, we found that ETC-1002-CoA was detected in liver, but not in skeletal muscle or in adipose tissue from ETC-1002-treated mice (Supplementary Table 1). Investigations in microsome preparation from human tissues demonstrated that while ACSVL1 was highly expressed in liver, it was only minimally detected in kidney, and was undetectable in skeletal muscle (Fig. 2e). In contrast to observations in liver microsomes, where ACSVL1 was highly expressed, the absence of ACSVL1 in human skeletal muscle corresponded to a lack of ETC-1002-CoA

synthetase activity (Fig. 2f); however, CoA activation of palmitate was still observed in human skeletal microsomes indicating the activity of other ACS isoforms was present (Fig. 2f). Importantly, consistent with the absence of ACSVL1 expression in primary human skeletal muscle myotubes (Fig. 2g), ETC-1002 did not suppress cholesterol synthesis (Fig. 2h) or induce signs of muscle apoptosis or cytotoxicity, a finding that was in stark contrast to that seen with simvastatin or atorvastatin (Fig. 2i,j and Supplementary Fig. 5). Similar observations were also made in rodent immortalized L6 myotubes (Supplementary Figs 2–4). Collectively, these findings indicate that the CoA activation of ETC-1002 and subsequent suppression of cholesterol synthesis requires ACSVL1, and since ACSVL1 is not expressed in skeletal muscle, ETC-1002 is unlikely to cause the associated myotoxicity.

**ETC-1002 reduces atherosclerosis independently of AMPKβ1.** Given that ACSVL1 is predominantly expressed in the liver and that modulation of both ACL and AMPK activity could affect blood levels of atherogenic lipoproteins, we next aimed to investigate the relative importance of these pathways for mediating the cholesterol-lowering effects of ETC-1002, and whether these effects lead to reduced vascular lesion development. To this end we crossed APOE-deficient ($Apoe^{-/-}$) mice with mice lacking AMPK β1 ($Ampkβ1^{-/-}$) to generate DKO mice. We have previously shown that AMPK β1 null mice have marked ($>90\%$) reductions in liver AMPK activity[49]; an effect also observed when crossed onto the APOE-deficient background as detected by significant reductions in activating phosphorylation at AMPK αT172 and phosphorylation of its downstream substrate acetyl-CoA carboxylase (ACC; Fig. 3a). Consistent with previous results, we observed that ETC-1002 increased liver AMPK activity in $Apoe^{-/-}$ mice, but this effect was not observed in DKO mice lacking AMPK β1 (Fig. 3a). We also confirmed ACSVL1 expression in liver using skeletal muscle as a negative control (Fig. 3b).

While mice fed a high-fat-high cholesterol (HFHC) diet displayed significant increases in body weight, adiposity, fasting glucose and diminished glucose tolerance compared with chow-fed mice, no statistically significant genotype or ETC-1002 ($30\,mg\,kg^{-1}$ per day) treatment effect on these parameters were observed (Supplementary Fig. 6A–H). HFHC feeding also increased very-low-density lipoprotein (Fig. 3c,d), LDL (Fig. 3c,e) and total cholesterol (Fig. 3c,g) by $>2.5$-fold in both $Apoe^{-/-}$ and DKO mice; however, no differences in plasma lipoprotein protein profiles were observed between genotypes. Importantly, ETC-1002 treatment reduced LDL-C in both $Apoe^{-/-}$ and DKO mice to a similar degree (38% and 44%, respectively; Fig. 3c,e), and had no effect on other lipoprotein fractions (Fig. 3c,d,f). Although ETC-1002 markedly reduced LDL-C, because $Apoe^{-/-}$ mice carry the majority of their plasma cholesterol in the very-low-density lipoprotein fraction, this effect corresponded to only a modest reduction in total plasma cholesterol of 18% in $Apoe^{-/-}$ mice and 13% in DKO, although the effect did not reach statistical significance in DKO mice (Fig. 3g). We then assessed the effects of HFHC feeding and ETC-1002 treatment on hepatic lipids and showed that HFHC feeding increased hepatic cholesterol mass by $\sim2$-fold in both $Apoe^{-/-}$ and DKO mice, an effect that was almost completely blocked by ETC-1002 treatment (Fig. 3h). HFHC feeding also increased hepatic triglycerides by $\sim2$-fold. Remarkably, ETC-1002 treatment completely prevented the increase in triglycerides in both genotypes, showing a 74% and 69% reduction in $Apoe^{-/-}$ and DKO mice, respectively (Fig. 3i).

Changes in the rates of liver lipogenesis and fatty acid oxidation are important determinants in controlling liver lipid content[50]. Consistent with decreased lipogenesis and increased fat oxidation, ETC-1002 treatment suppressed the respiratory exchange ratio (RER) during the dark (feeding) cycle in both $Apoe^{-/-}$ and DKO mice (Fig. 3j,k), while no treatment effect was observed during the light (fasted) cycle (Fig. 3j,l). To further investigate the mechanisms contributing to AMPK-dependent and -independent effects on lipid metabolism *in vivo*, we conducted experiments in primary mouse hepatocytes from AMPK β1 knockout (KO) mice. We found that consistent with the AMPK-independent reductions in liver lipids and RER observed *in vivo*, ETC-1002 suppressed total lipid synthesis in both wild-type and AMPK β1 KO hepatocytes (Fig. 3m). These data indicate that ETC-1002 reduces LDL-C, as well as liver triglycerides and cholesterol through AMPK-independent pathways.

We then investigated the mechanism by which ETC-1002 reduces LDL-C. It has been shown that statins reduce LDL-C by triggering a well-characterized feedback mechanism whereby inhibition of cholesterol synthesis results in reduced cellular cholesterol levels, which activates sterol response element-binding protein-2 (SREBP2)-dependent LDL receptor transcription. Upregulation of the LDL receptor results in a new homeostatic state, where more cellular cholesterol is derived from blood LDL particles, thus reducing blood LDL-C[51–53]. Given the effects of ETC-1002 treatment on lipid synthesis and hepatic cholesterol, we investigated whether ETC-1002 treatment also increased hepatic SREBP-dependent gene expression, and whether this was affected by the absence of AMPK β1. Indeed, AMPK signalling has been shown to regulate some SREBP isoforms through phosphorylation[32]. As expected, HFHC feeding significantly suppressed the expression of numerous genes in $Apoe^{-/-}$ mice known to be under SREBP2-dependent transcriptional regulation, including *Srebf2*, *Ldlr*, *Pcsk9* and *Hmgr*, while a slight increase in *Srebf1c* expression was observed (Supplementary Fig. 6I). Interestingly, *Srebf2* expression was elevated in DKO mice compared with $Apoe^{-/-}$ mice; however, consistent with the absence of a genotype effect on plasma LDL-C levels, this did not result in changes in *Ldlr* expression. No differences in *Acly* expression between genotypes were observed. Importantly, ETC-1002 treatment increased *Srebf2* and *Ldlr* expression by $>2$-fold in both $Apoe^{-/-}$ and DKO mice when compared with their respective HFHC-fed controls (Fig. 3n), an effect that was further supported by an increase in plasma membrane-associated LDLR in liver sections from ETC-1002-treated mice (Fig. 3o). ETC-1002 also increased *Pcsk9*, *Acly* and *Srebf1c* in both genotypes, although this effect did not achieve statistical significance in DKO mice. Similar to *Srebf2*, *Slc27a2* (ACSVL1) expression was increased by ETC-1002 treatment and in DKO mice compared with $Apoe^{-/-}$ mice (Fig. 3n), suggesting that *Slc27a2* may also be subject to transcriptional regulation by SREBP2. Although some ACS isoforms have been shown to be regulated by SREBP2 (ref. 54), other effects on lipid metabolism resulting from loss of AMPK β1 or ETC-1002 treatment cannot be ruled out. These findings in $Apoe^{-/-}$/AMPK β1 DKO mice and primary hepatocytes demonstrate that the mechanism by which ETC-1002 lowers LDL-C is consistent with compensatory SREBP2- dependent LDL receptor upregulation in response to AMPK-independent suppression of cholesterol synthesis.

We then evaluated whether the reductions in plasma cholesterol observed with ETC-1002 treatment translated to reductions in vascular lesion size. Morphological assessments of lesions from haematoxylin and eosin-stained sections from the aortic sinus of these mice revealed attenuated lesion development in chow-fed $Apoe^{-/-}$ mice, whereas HFHC-fed $Apoe^{-/-}$ and DKO mice developed significantly larger lesions (Fig. 4a). However, no difference in lesion size was observed between $Apoe^{-/-}$ and DKO mice fed a HFHC diet suggesting that the

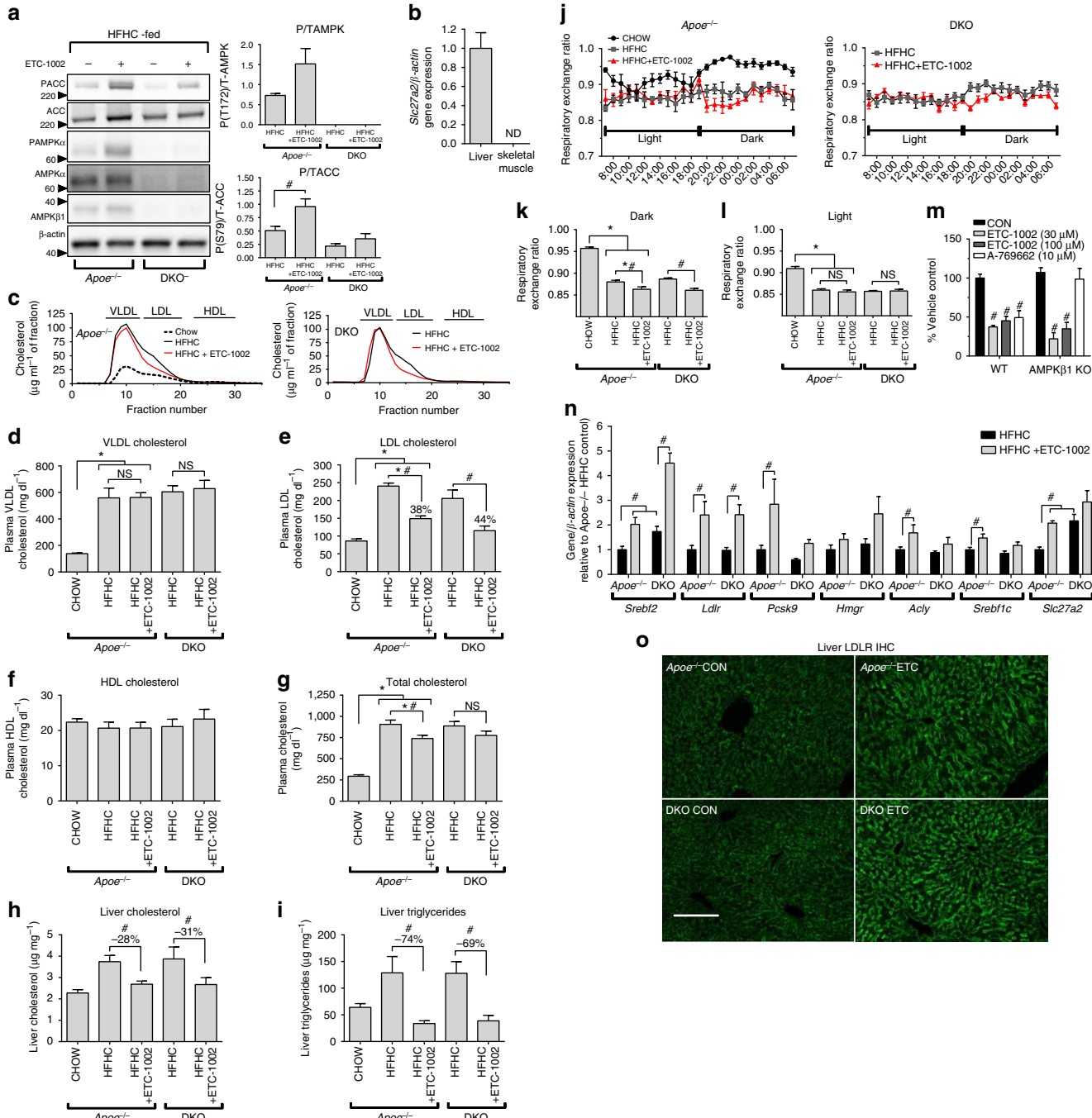

**Figure 3 | ETC-1002 mediates effects on lipid metabolism independently of AMPK β1.** $Apoe^{-/-}$ and $Apoe^{-/-}/Ampk\beta1^{-/-}$ (DKO) mice were fed a HFHC diet for 12 weeks with or without ETC-1002 targeted to achieve a $30\,mg\,kg^{-1}$ per day dose and (**a**) total and phosphorylated (T172) AMPKα and total and phosphorylated (S79) ACC measured in livers from $Apoe^{-/-}$ ($n = 4$) and DKO ($n = 3$). (**b**) Slc27a2 (ACSVL1) expression in liver and tibialis anterior from untreated $Apoe^{-/-}$ mice ($n = 6$). (**c**) Representative fast performance liquid chromatography tracings from chow- and HFHC diet-fed ± ETC-1002 $Apoe^{-/-}$ and DKO mice. Plasma (**d**) very-low-density lipoprotein (VLDL), (**e**) LDL, (**f**) HDL ($n = 4$ for chow-fed mice and 6 for remaining treatment groups) and (**g**) total cholesterol ($n = 10$), liver (**h**) cholesterol and (**i**) triglycerides ($n = 4$ for chow-fed and 6 for remaining treatment groups) determined at the end of study. (**j**) RER was measured over 24 h following 10 weeks on diet and mean RER during (**k**) dark (19:00–07:00) and (**l**) light (07:00–19:00) cycles calculated. (**m**) Total lipid synthesis in primary hepatocytes isolated from WT and AMPK β1 KO mice treated for 4 h with the indicated concentration of ETC-1002, A-769622 or vehicle (CON) in the presence of $^3$H-acetate ($n = 3$–8 independent experiments performed in duplicate). (**n**) Srebf2, Ldlr, Pcsk9, Hmgr, Acly, Srebf1c and Slc27a2 gene expression ($n = 4$ chow-fed and 5 for remaining treatment groups) determined at the end of study. (**o**) Representative images for immunohistochemistry staining of LDLR in frozen liver sections from ETC-1002-treated $Apoe^{-/-}$ and DKO mice (20 × ); scale bar, 100 μM. Data are expressed as mean ± s.e.m. Multiple comparisons were made using an *one-way (within $Apoe^{-/-}$ treatment groups) or #two-way ANOVA (between $Apoe^{-/-}$ and DKO treatment groups); Bonferroni's multiple comparisons test; *and #$P < 0.05$.

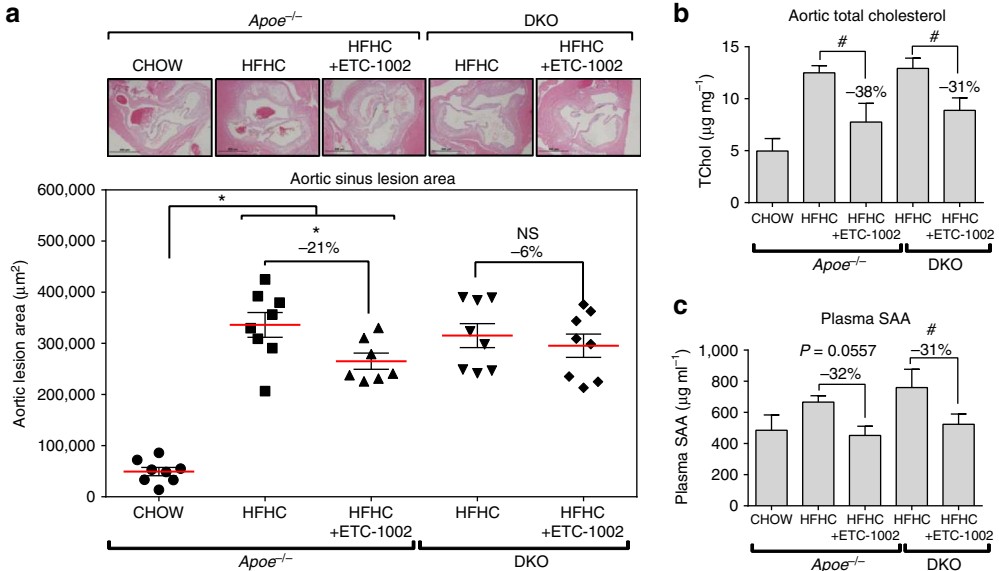

**Figure 4 | ETC-1002 treatment reduces the progression of atherosclerosis.** $Apoe^{-/-}$ and $Apoe^{-/-}/Ampk\beta1^{-/-}$ (DKO) mice were fed a HFHC diet for 12 weeks with or without ETC-1002 targeted to achieve a 30 mg kg$^{-1}$ per day dose. (**a**) Sections of the aortic sinus from control and ETC-1002-treated mice were stained with haematoxylin and eosin and atherosclerotic lesion size determined; $n = 8$ for all treatment groups except ETC-1002-treated $Apoe^{-/-}$ ($n = 7$). (**b**) Total cholesterol (TChol) was measured in whole aorta ($n = 4$ for $Apoe^{-/-}$ and 6 for DKO HFHC-fed mice) and (**c**) plasma SAA measured at the end of study ($n = 10$). Data are expressed as mean ± s.e.m. Multiple comparisons were made using an *one-way (within $Apoe^{-/-}$ treatment groups) or #two-way ANOVA (between $Apoe^{-/-}$ and DKO treatment groups); Bonferroni's multiple comparisons test; *and #$P < 0.05$. NS, not significant.

absence of AMPK β1 did not accelerate atherosclerosis. Despite the modest reductions in total plasma cholesterol by ETC-1002 treatment, quantitation of sections from the aortic sinus of ETC-1002-treated $Apoe^{-/-}$ mice showed a marked reduction (21%; $P < 0.05$, one-way analysis of variance (ANOVA) and Bonferroni's multiple comparisons test) in lesion size (Fig. 4a). Similar to our observations on plasma total cholesterol, aortic lesions from ETC-1002-treated DKO mice trended lower than the DKO control group; however, the reduction was attenuated compared with $Apoe^{-/-}$ mice and did not reach statistical significance. Analysis of whole aorta cholesterol levels showed that increases in response to HFHC feeding ($>2$-fold) was markedly attenuated by ETC-1002 treatment in both $Apoe^{-/-}$ ($-38\%$) and DKO ($-31\%$) mice (Fig. 4b), suggesting AMPK independence. Investigations in bone marrow-derived macrophages from wild-type and AMPK β1 KO mice showed that consistent with the absence of ACSVL1 expression (Supplementary Fig. 7A), ETC-1002 did not activate AMPK (Supplementary Fig. 7B) nor affect *de novo* lipid synthesis rates in macrophages (Supplementary Fig. 7C) suggesting that the effects of ETC-1002 are not mediated by direct effects on lesion macrophages. To assess whether the effects of ETC-1002 on hepatic lipid metabolism was associated with reductions in low-grade systemic inflammatory status, we measured plasma serum amyloid A (SAA), a liver-derived acute phase protein that has been shown to be elevated by HFHC feeding. Plasma SAA levels were reduced by $>30\%$ by ETC-1002 treatment in both $Apoe^{-/-}$ and DKO mice, indicating a reduction in diet-induced low-grade inflammation (Fig. 4c). These findings suggest that the anti-atherosclerotic effects of ETC-1002 are primarily mediated by AMPK-independent effects on hepatic lipid metabolism resulting in reduced low-grade inflammation and plasma levels of atherogenic LDL-C.

**ETC-1002 upregulates LDLR activity in human hepatocytes.** To examine whether the effects on LDL metabolism observed in mice

were linked to cholesterol synthesis inhibition in humans, we treated primary human hepatocytes (PHH) with ETC-1002 and found that it reduced the incorporation of [$^3$H]-H$_2$O into sterol and fatty acid fractions by 59% and 50%, respectively (Fig. 5a). Concentration response studies demonstrated that the half maximal effect concentration for inhibition of *de novo* cholesterol synthesis was $\sim10\,\mu$M (Fig. 5a). Following a 36 h treatment, both ETC-1002 and atorvastatin reduced total intracellular cholesterol mass by 21% and 42%, respectively (Fig. 5b), which corresponded to a 31% and 32% reduction in media apoB concentrations (Fig. 5c). Assessments of media apoB levels at earlier time points demonstrated that ETC-1002 did not affect secretion rates (Fig. 5d), which was supported by a lack of an effect on [$^{14}$C]-oleate-derived triglyceride secretion (Fig. 5e). However, a significant reduction in media apoB levels was observed following 24 h of treatment, suggesting a potential increase in LDL receptor-mediated apoB uptake. To further validate this observation, we first measured whether ETC-1002 treatment increased the expression of sterol-responsive genes consistent with activation of SREBP2-dependent gene transcription in humans as observed in mice *in vivo*. Similar to atorvastatin treatment, ETC-1002 increased the expression of *SREBF2*, *HMGR*, *PCSK9* and *LDLR* mRNA, with maximum effects reaching 1.4, 1.3, 1.5 and 2.3-fold, respectively (Fig. 5f). Furthermore, ETC-1002 and atorvastatin treatment also increased DiI (3,3′-dioctadecylindocarbocyanine iodide)–LDL association, confirming that like statins, ETC-1002 treatment upregulates LDLR activity in human liver cells. Interestingly, A-769662 was inactive, suggesting that β1-selective AMPK activation was not sufficient to increase LDLR activity in PHH. These findings further implicate the role of ACL inhibition in the LDL-C-lowering effects of ETC-1002 (Fig. 5g).

**ACL suppression increases LDLR activity.** As we had shown that the lipid-lowering effects of ETC-1002 were AMPK independent, we finally wanted to establish whether suppression of

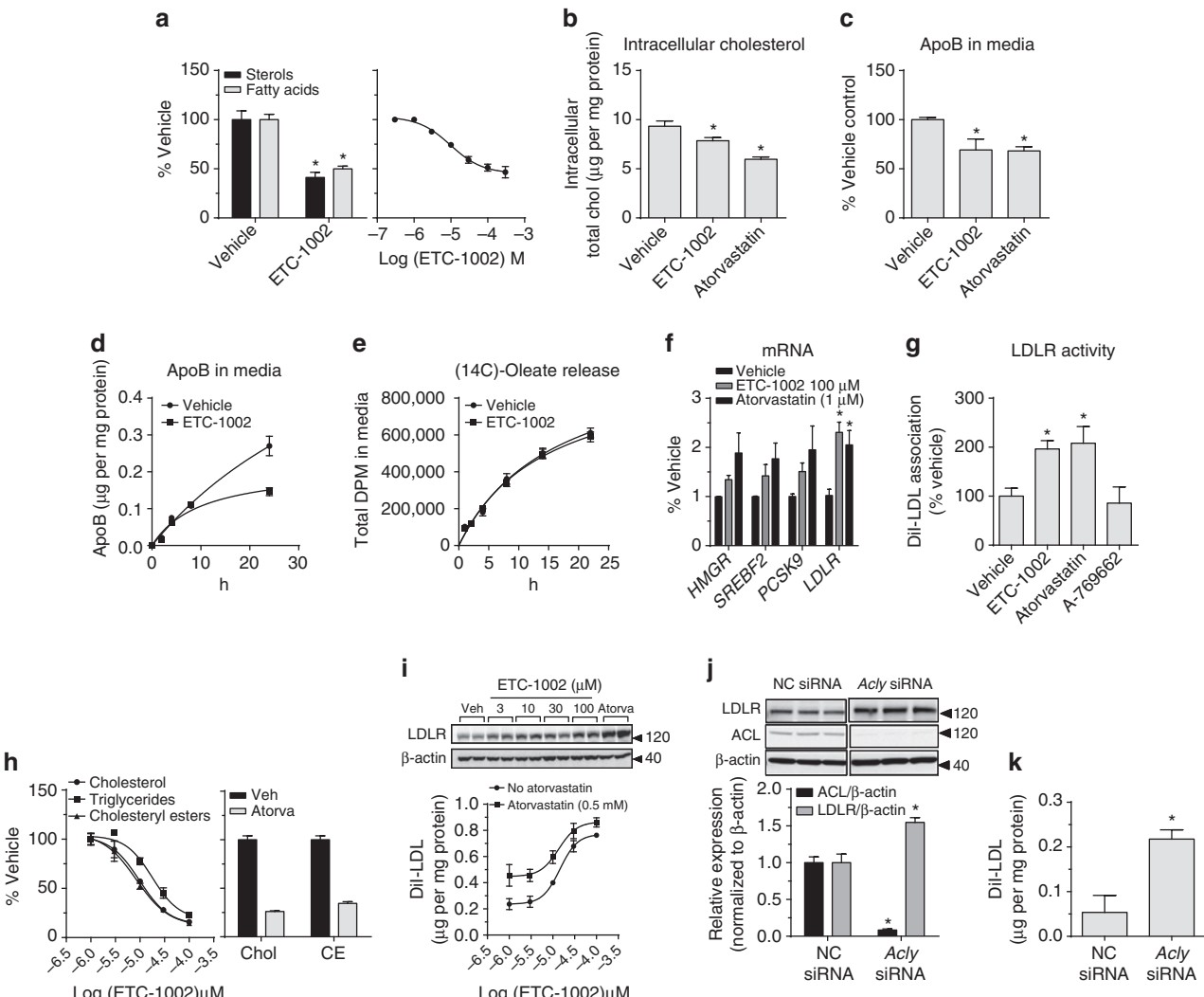

**Figure 5 | ACL suppression increases LDLR.** (**a**) Primary human hepatocytes were treated with vehicle or ETC-1002 (100 μM) for 18 h in the presence of [³H]-H₂O or pretreated with the indicated concentrations of ETC-1002 for 1h, following a 3 h [¹⁴C]-acetate pulse. Counts incorporated into the non-saponifiable (sterols), and saponifiable (fatty acids) lipid fractions were determined and expressed as % vehicle. (**b**) Primary human hepatocytes were treated with vehicle, atorvastatin (0.5 μM) or ETC-1002 (100 μM) for 36 h and total intracellular cholesterol concentrations were determined in cell lysates. (**c**) Media was assayed for apoB by enzyme-linked immunosorbent assay. (**d**) PHHs were treated with ETC-1002 for the indicated time points and the apoB concentrations were measured in the media. (**e**) PHH were pre-labelled overnight with [¹⁴C]-oleate and the effects of ETC-1002 treatment on oleate-derived counts in the media determined at the indicated time points. PHH hepatocytes were treated with ETC-1002 and atorvastatin, and (**f**) mRNA levels for *HMGR*, *SREBF2*, *PCSK9* and *LDLR* were determined by reverse transcription–quantitative PCR or (**g**) LDL receptor activity determined; A-769662 (10 μM). (**h**) RH7777 cells were treated with the indicated concentrations of ETC-1002 or atorvastatin (0.5 μM), and [¹⁴C]-lactate incorporation into cholesterol (Chol), cholesteryl esters (CE) and triglycerides determined. (**i**) LDLR-mediated DiI–LDL uptake in response to the indicated concentrations of ETC-1002 ± atorvastatin (0.5 μM). RH7777 cells were subjected to negative control (NC) or *Acly* siRNA-mediated gene silencing (n = 3) and analysed for (**j**) ACL LDLR protein levels, or (**k**) LDLR-mediated DiI–LDL uptake. Data for PHH are expressed as mean ± s.e.m. of n = 2 donors performed in triplicate (**a–c,f,g**; except A-769662 treatment was 1 donor) or representative of two independent experiments showing similar results (**d,e**). (**h**) IC₅₀ and (**i**) EC₅₀ determinations were made using nonlinear curve fit model. Multiple comparisons were made using (**a,j,k**) unpaired Student's *t*-test or (**b,c,f,g**) an one-way ANOVA, Bonferroni's multiple comparisons test; *$P < 0.05$.

ACL activity by ETC-1002 was sufficient to trigger LDL receptor upregulation. We first demonstrated model suitability by confirming that our hepatocytes (McArdle cells-RH7777) were sensitive to pharmacological inhibition of cholesterol synthesis activity by ETC-1002 and atorvastatin treatment. Consistent with our observations in both primary mouse (Fig. 3l) and human hepatocytes (Fig. 5a), ETC-1002 elicited concentration-dependent inhibition of [¹⁴C]-lactate incorporation into cholesterol, cholesteryl ester and triglycerides (IC₅₀ = 9.7, 8.4 and 17.8 μm, respectively; Fig. 5h). Atorvastatin treatment also decreased cholesterol and cholesteryl ester synthesis (Fig. 5h). These effects

of ETC-1002 treatment were associated with a 3.2-fold (EC₅₀ = 15 μM) increase in LDL receptor activity (Fig. 5i), and in support of our clinical findings, the addition of ETC-1002 to atorvastatin increased LDL receptor activity above atorvastatin treatment alone[29], supporting that the co-suppression of ACL and HMG-CoA reductase activity is complementary (Fig. 5i). Transfection of hepatocytes with *Acly* siRNA resulted in an 80% reduction in ACL protein (Fig. 5j), which corresponded to more than a 50% increase in LDL receptor protein (Fig. 5j) and a 4-fold increase in LDL receptor activity (Fig. 5k). These findings support the critical requirement of ACL activity to supply substrate for

cholesterol biosynthesis and demonstrate an important regulatory link between ACL activity and LDL receptor regulation, thus demonstrating that ACL inhibition is a novel molecular target to reduce LDL-C.

## Discussion

ETC-1002 is a small-molecule cholesterol synthesis inhibitor being developed to lower elevated levels of LDL-C that has been previously shown to inhibit hepatic ACL and promote AMPK signalling. We have further investigated the underlying mechanism for these activities and demonstrate the requirement for CoA activation of ETC-1002 to directly modulate the activities of both enzymes. Importantly, we identify ACSVL1 as the specific ACS isoform responsible for catalysing the CoA activation of ETC-1002 and demonstrate the requirement of this activity to directly inhibit ACL and mediate β1-dependent AMPK activation. Owing to the known tissue expression profile of ACSVL1, these findings demonstrate that modulation of ACL and AMPK activities by ETC-1002-CoA is almost exclusively restricted to liver. We then extrapolate these findings to provide a mechanistic basis for differentiation from the myotoxic effects of statins by showing that the absence of ACSVL1 expression in human skeletal muscle precludes ETC-1002 from inhibiting the cholesterol biosynthesis pathway, a critical source of important biological intermediates essential to maintain normal muscle cell function[14–18].

Using PHH and a novel $Apoe^{-/-}/Ampk\ \beta1^{-/-}$ DKO mouse model, we exclude the involvement of AMPK signalling in the

mechanism for LDL-C lowering by ETC-1002 and establish ACL as the molecular target. ACL is a cytosolic enzyme upstream from HMG-CoA reductase that catalyses the first committed step in the utilization of citrate derived from the mitochondrial oxidation of carbohydrates for lipid synthesis. Suppression of ACL activity leads to a reduction in cytosolic acetyl-CoA, a required precursor for cholesterol and fatty acid synthesis[26], and subsequent compensatory LDL receptor upregulation[20,24,25,55,56]. We show that similar to statins, inhibition of ACL by ETC-1002 leads to reduced cholesterol biosynthesis and the upregulation of LDL receptor expression in PHH, and $Apoe^{-/-}$ and DKO mice in vivo. Furthermore, we demonstrate the therapeutic utility of this mechanism by showing that the resulting reductions in plasma LDL-C were associated with a proportionate reduction in atherosclerosis. ETC-1002 treatment markedly reduced whole aortic cholesterol and lesion size within the aortic sinus. The absence of ACSVL1 expression in macrophages precludes the contribution of direct effects of ETC-1002 on lesion macrophages and suggests that its anti-atherosclerotic activity is primarily driven by reduced systemic inflammation and LDL-C secondary to inhibition of hepatic ACL.

Pharmacological inhibition of cholesterol biosynthesis leads to effective lowering of elevated LDL-C, a validated biomarker for ASCVD risk reduction in hyperlipidemic patients[3–5]. Despite the proven benefits of statins, many patients remain at risk for cardiovascular disease due to muscle-related adverse effects, which prevent them from tolerating a statin dose required to achieve recommended LDL-C goals. As such, the management of LDL-C levels can also be achieved by other mechanisms such as

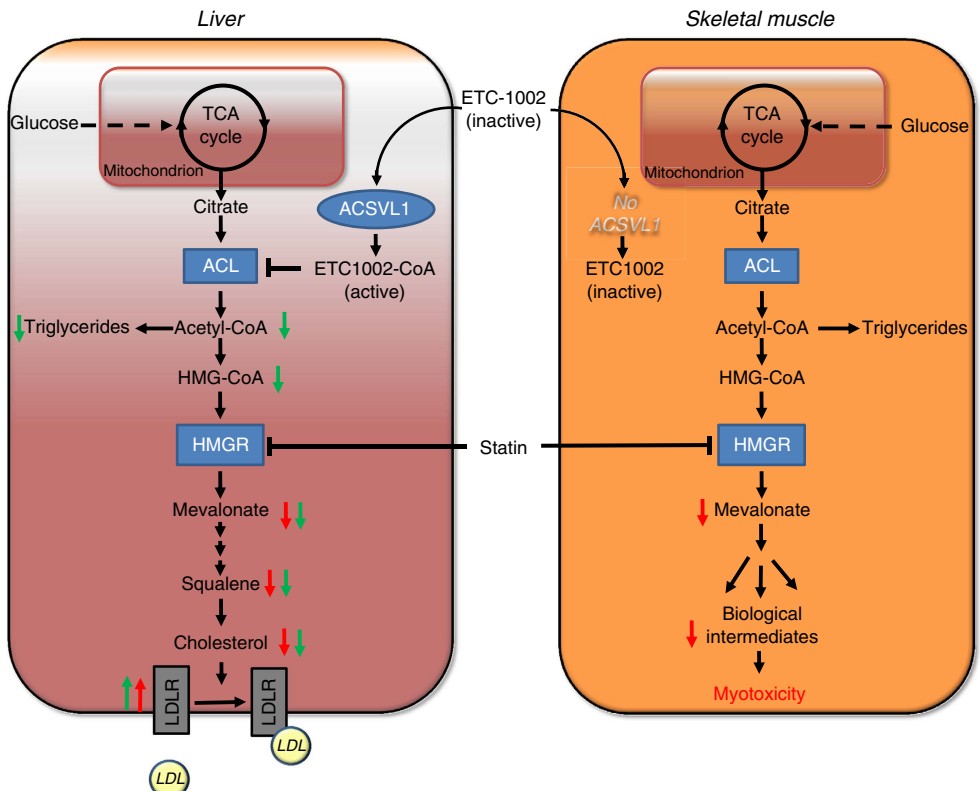

**Figure 6 | The mechanism of action of ETC-1002.** In liver, ETC-1002 (bempedoic acid) is activated to ETC-1002-CoA by ACSVL1, and subsequently inhibits ACL. Similar to inhibition of HMG-CoA reductase (HMGR) by statins, inhibition of liver ACL by ETC-1002-CoA results in the suppression of cholesterol synthesis and compensatory LDLR upregulation and LDL particle clearance from the blood. Skeletal muscle does not express ACSVL1 and is unable to convert ETC-1002 to its active form. Therefore, ETC-1002 does not suppress the synthesis of cholesterol or the associated biological intermediates that are required to maintain normal muscle cell function, or promote the associated toxicity. Green and red arrows indicate the effects of ETC-1002 and statins, respectively.

inhibition of cholesterol absorption in the gut (that is, ezetimibe), or preventing LDL-receptor degradation (that is, PCSK9 inhibitors)[57–59]. Importantly, each of these mechanisms primarily reduces LDL-C by upregulating the activity of the LDLR, a mechanism proven to reduce cardiovascular events[60–62]. These strategies have significantly influenced further expectations for cardiovascular risk reduction in patients with high-risk co-morbidities, which has led to the combination of statins with other existing LDL-C-lowering agents and the need for novel therapies with mechanisms that complement the effects of statins without increasing adverse effects in skeletal muscle[57–59]. Our findings establish ACL as a viable target for therapeutic intervention by reducing hepatic cholesterol synthesis and increasing LDL receptor activity. Furthermore, characterization of the underlying mechanism leading to the CoA activation of the prodrug, ETC-1002, provides a mechanistic basis for differentiation from statin therapy by improving the liver specificity for cholesterol biosynthesis inhibition and thereby decreasing the potential for muscle-related adverse effects (Fig. 6).

## Methods

**Materials.** Dulbecco's modified Eagle media (DMEM), non-essential amino acids, HEPES, phosphate-buffered saline (PBS), sodium pyruvate, penicillin/streptomycin and DiI–LDL were obtained from Invitrogen (Logan, Utah). Fetal bovine serum (FBS) was obtained from Hyclone (Grand Island, New York). Bovine albumin, fraction V, insulin, hydrocortisone, simvastatin and atorvastatin were acquired from Sigma Chemical Company (St Louis, MO). Succinic acid, octadecanedioic acid and arachidic acid were obtained from TCI Chemical (Portland, OR). All remaining fatty acids were obtained from Sigma Chemical Company (St Louis, MO). Triacsin C, pioglitazone, rosiglitazone, troglitazone and ACSVL1 antibody were obtained from Abcam (Cambridge, MA). Radiochemicals [$^3$H]-H$_2$O, [(U)$^{14}$C]-glucose [1-$^{14}$C]-acetic acid, [$^{14}$C]-citrate and [$^{14}$C]-oleate were obtained from American Radiolabeled Chemicals, Inc. (St Louis, MO) or PerkinElmer (Waltham, MA). Biocoat type I collagen-coated plates were purchased from Becton Dickinson Labware (Bedford, MA). Primary antibodies to P/T(23A3), AMPK (#2531, #2603), P/T(C83B10), ACC (#3661, #3676), AMPKβ1 (#12063), ACL (#4332) and β-actin(13E5) (#4970) were diluted 1:1,000, and secondary antibody (#7074) was diluted 1:10,000, and were all obtained from Cell Signaling Technologies (Beverly, MA). Primary antibodies to LDLR(EP1553Y) (#Ab52818) diluted 1:1,000 and ACSVL1 diluted to 1 µg ml$^{-1}$ (#ab83763) were purchased from Abcam. ApoB and ApoAI ELISA were obtained from ALerChek (Springvale, Main). HPLC grade reagents, solvents and Ultima Gold scintillation cocktail were obtained from Sigma-Aldrich (St Louis, MO). Microscint O was purchased from PerkinElmer (Waltham, MA).

**ETC-1002 formulation.** For in vitro assays, ETC-1002 was formulated using aseptic technique at 30 and 100 mM in sterile dimethylsulphoxide and stored in sterile microcentrifuge tubes at 4 °C for up to 4 weeks. Working solutions of ETC-1002 were prepared in serum-free medium containing 12 mM HEPES, 10,000 U ml$^{-1}$ penicillin and 100 µg ml$^{-1}$ streptomycin.

**Cell culture and siRNA transfection.** Authenticated McA-RH7777 cells were purchased from ATCC (CRL-1601) free of mycoplasma and used for no more than three passages. Cells were grown and treated in DMEM containing 1 g l$^{-1}$ D glucose, supplemented with 10% FBS. Reverse transfections were performed in six-well culture plates at $2.5 \times 10^5$ cells per well using Lipofectamine 2000 Invitrogen. Cells were incubated for 48 h with 10 nM silencer siRNA for ACSVL1 or ACL or negative control Invitrogen.

**Primary human myotubes.** Cryopreserved primary human myocytes (Lonza CC-2561) were thawed, centrifuged and resuspended in complete media (SkGM-2 Bullet kit, Lonza) following the manufacturer's directions. Cells were plated in Laminin-coated plates (Corning) and allowed to adhere and grow to 90% confluence. Media was changed to basal media containing 2% horse serum (Life Technologies, Inc.) and 5 µg ml$^{-1}$ insulin, and cells continued in culture until myotubes formed. For sterol synthesis assays, fresh basal media containing [$^{14}$C]-glucose, and vehicle, ETC-1002, simvastatin, or atorvastatin was added to myotube cultures for an additional 12 h. Viability after 12 h was assessed in parallel cultures using MTT assay, and non-saponifiable lipids were extracted as described below. Morphology was determined by visual assessment of merged images from bright-field and fluorescent capture ($\times$20) of Hoechst-stained cultured primary human myotubes treated with vehicle, statins ($\pm$500 µM mevalonate) or ETC-1002 (100 µM) for 48 h. Viability/cytotoxicity was determined after 48 h treatment by

measuring GF-AFC/bis-AAF-R110 cleavage and Caspase 3, 7 activity luciferase-based DEVD cleavage (ApoTox-Clo, Promega).

**Primary human hepatocytes.** Cryopreserved PHH (Triangle Research Labs, LLC, Charlottesville, VA) were thawed, centrifuged and resuspended in complete William's E medium containing 10% FBS. Cells were plated in collagen-coated plates and allowed to adhere for 4 h. For [$^3$H] H$_2$O incorporation studies, cells were switched to serum-free medium and incubated for an additional 18–20 h in the presence of [$^3$H]-H$_2$O $\pm$ indicated compounds before lipid extraction. For [1-$^{14}$C]-acetate incorporation studies, cells incubated overnight in serum-free medium. Cells were washed and preincubated with fresh medium $\pm$ compounds for 1 h before the addition of [$^{14}$C]-acetate. Cells incubated for an additional 3 h before lipid extraction described below.

**De novo lipid synthesis assay.** Rates of lipid synthesis were assessed in cultured McArdle hepatoma cells, PHH and skeletal muscle myotubes using [(U)$^{14}$C]-lactate, [1-$^{14}$C]-acetate or [$^3$H]-H$_2$O. Experiments were performed in DMEM with 4.5 g l$^{-1}$ glucose. Cells were treated with compound or vehicle (0.1% dimethylsulphoxide) for up to 4 h followed by lipid isolation. After metabolic labelling, cell extracts were processed for thin-layer chromatography, or saponified and non-saponified lipids were extracted from cells. Briefly, cells were scraped in 1 M KOH/EtOH and transferred to glass vials. Wells were washed with 1 M KOH/EtOH and added to the same glass vial. Samples were vortexed and heated for 2 h with occasional vortexing, and then cooled to room temperature. For the isolation of the sterol fraction, 1 part H$_2$O and 2 parts n-hexanes were added per sample, and vials were capped, vortexed and centrifuged for 5 min at room temperature. The top organic layer was transferred to a new tube and sample was extracted again as above in 2 parts n-hexanes. Each sample was heated to 50 °C under N$_2$ gas until all solvent was evaporated. Samples were re-suspended in toluene and diluted in scintillant. For the fatty acid fraction, 1 part 2 N HCL and 2 parts petroleum ether was added to the remaining aqueous layer of each sample. Samples were extracted twice in petroleum ether as described above and transferred to a new glass vial. Samples were heated to 50 °C under N$_2$ gas until all solvent was evaporated. Samples were re-suspended in toluene and diluted in scinillant.

**AMPK activity assay.** AMPK activity was determined by measuring the phosphorylation of the ULight-SAMS peptide (sequence: CHMRSAMSGLHLVKRR synthetic peptide derived from residues 73–85 of rat acetyl-CoA carboxylase in which Ser77 was mutated to Ala; phosphorylation site: Ser79; #TRF0208, PerkinElmer) using time-resolved fluorescence resonance energy transfer. Briefly, 0.5 nM active recombinant full-length human AMPK heterotrimers (isolated from Sf9 cells obtained from Sigma-Aldrich) was pretreated with the indicated activators in 30 µl kinase buffer containing 50 mM HEPES (pH 7.5), 1 mM EGTA, 2 mM dithiothreitol, 0.01% Tween, in white opaque 96-well microplates at 37 °C for 15 min. Reactions were returned to room temperature on an orbital plate shaker for 5 min before a 10 µl addition of a mixture containing $4\times$ ATP (30 µM final) and Ulight SAMS (50 nM final). Plates were briefly centrifuged at 2,000 r.p.m. and placed back on the plate shaker at room temperature for 15 min. Reactions were stopped by the addition of 40 µl of detection mix (CR97–100, PerkinElmer) containing 40 mM EDTA and 8 nM Eu-anti-phospho ACC antibody (#TRF0118-D, PerkinElmer). SAMS phosphorylation was determined by time-resolved fluorescence resonance energy transfer (Lm1 Ex = 330 nm, Em = 668 nm (630 nm Co); Lm2 Ex = 330 nm, Em = 620 nm (570 nm Co). The 668/620 nm fluorescence emission ratio was calibrated to standardized active AMPKα1β1γ1 enzyme with reported activity of 685 nmole min$^{-1}$ mg$^{-1}$.

**ACL enzyme activity assay.** The activity of recombinant human ACL was carried out essentially as described in ref. 63. Briefly, $7.5\times$ compounds were added to a 96-well PolyPlate containing 60 µl of buffer (87 mM Tris, pH 8.0, 20 µM MgCl$_2$, 10 mM KCl and 10 mM DTT) per well with substrates CoA (200 µM), ATP (400 µM) and [14C]-citrate (specific activity: 2 µCi µmol$^{-1}$)(150 µM). Reaction was started with 4 µl (300 ng per well) ACL and the plate incubated at 37 °C for 3 h. The reaction was terminated by the addition of 3.5 µl 500 mM EDTA. 200 µl MicroScint O was then added to the reaction mixture and incubated at room temperature overnight with gentle shaking. The [$^{14}$C] acetyl-CoA signal was detected (5 min per well) in a TopCount NXT liquid scintillation counter (Perkin-Elmer, Waltham, MA).

**LDLR activity assay.** Cells were seeded in 12-well collagen-coated plates at $\sim$60% confluence in complete DMEM 10% FBS and allowed to grow overnight. Cells were switched to 5% FBS media $\pm$ compounds and incubated overnight. Cells were washed and serum-free DMEM 0.2% FA-free BSA was added $\pm$ compound and 10 µg ml$^{-1}$ DiI–LDL. Cells incubated between 2 and 6 h before placed on ice and extensively washed with ice-cold PBS. Cholesterol was extracted into 500 µl iso-propyl alcohol (IPA) for 15 min on plate shaker, transferred to an Eppendorf tube and centrifuged for 5 min at 10,000 g. A volume of 300 µl of each sample was transferred to a black 96-well plate and fluorescence measured. Standard DiI–LDL solutions were prepared in IPA, and µg DiI–LDL per mg cell protein was

calculated. LDLR-specific LDL association was determined by subtracting fluorescence from cells treated with LDLR-neutralizing antibody.

**ETC-1002-CoA synthetase activity assay.** Human liver subcellular fractions and pooled ($n = 50$) microsomes were obtained from XenoTech, and individual human donor microsomes were obtained from LifeTech. A volume of 25 μl of [$^{14}$C]-ETC-1002 (20 ×; 10 μM final) and 25 μl blank or competitive (20 ×) substrate was added to 425 μl 1 × ACS buffer containing 175 mM Tris HCl, 0.1% triton, 32 mM MgCl, 20 mM DTT, 40 mM ATP and 4 mM coenzyme in each reaction vial. Vials were placed on oscillating 37 °C water bath for 15 min before the addition of 25 μl (50 μg) of microsomes to each reaction vial. Following a 15 min incubation, reactions were stopped by the addition of 200 μl of 2 N HCl to each reaction vial. ETC-1002-CoA was separated by 4, 2 ml diethyl ether extractions and 150 μl aqueous from each reaction mixture was added to 5 ml of scintillation fluid and DPM determined.

**Reverse transcription–quantitative PCR.** Cryopreserved PHH (Triangle Research Labs) were seeded in 6-well collagen-coated plates and allowed to attach for 4 h. Cells were exposed to compounds at the indicated concentration overnight. Cells or mouse liver tissue were lysed in TRIzol reagent (ThermoFisher) to remove lipid, and the aqueous phase was applied to an RNeasy kit (Qiagen, CA, USA) column for subsequent purification. Relative gene expression was calculated using the comparative Ct ($2 - \Delta\Delta Ct$) method, where values were normalized to a housekeeping gene. Taqman primers *Slc27a2* (Mm00449517-m1), *Ldlr* (Mm01177349-m1), *Hmgr* (Mm01282499-m1), *Pcsk9* (Mm01263610-m1), *Acly* (Mm01302282-m1) and *Srebf2* (Mm01306292-m1) were purchased from ThermoFisher.

**Western blots.** Hepatocyte or RH7777 cell lysates were prepared using ~150–400 μl 1 × lysis buffer containing 20 mM Tris-HCl (pH 7.5), 150 mM NaCl, 1 mM Na₂EDTA, 1 mM EGTA, 1% Triton, 2.5 mM sodium pyrophosphate, 1 mM β glycerophosphate, 1 mM Na₃VO₄, 1 μg ml$^{-1}$ leupeptin, 1 mM phenylmethylsulphonyl fluoride and 1 × phosphatase inhibitor cocktail (Sigma). Total lysate protein concentrations were determined using the BCA Protein Assay (BioRad Laboratories, Hercules, CA). Protein concentrations were adjusted and diluted in 4 × LDS (lithium dodecyl sulphate gel sample buffer) containing 50 mM DTT. Proteins were separated using SDS–polyacrylamide gel electrophoresis (4–12%) Bis/Tris, MOPS running buffer (Invitrogen). Separated proteins were electrophoretically transferred to polyvinyl difluoride membranes. Non-specific binding was blocked and membranes were probed with antibodies against β-actin, total and phosphorylated forms of AMPK (PT722), and ACC (S79), and LDLR, ACL, ACSVL1 and AMPK β1. Cropped gel images are shown in main figures and uncropped gels are shown in Supplementary Fig. 8.

**Animals.** All animal procedures were approved by the McMaster University Animal Ethics Research Board (AUP #:12-12-44).

**$Apoe^{-/-}$/$Ampk\beta 1^{-/-}$ DKO mice.** $Apoe^{-/-}$ mice on a C57Bl6 background were purchased from JAX and crossed to $Ampk\beta 1^{-/-}$ mice that were generated on a C57Bl6 background[49]. Progeny were then crossed to generate $Apoe^{-/-}$ or $Apoe^{-/-}$/$Ampk\beta 1^{-/-}$ (DKO) mice. Male $Apoe^{-/-}$ and $Apoe^{-/-}$/$Ampk\beta 1^{-/-}$ mice were maintained on a 12 h light dark cycle (lights on at 07:00) and housed in a pathogen-free facility at 23 °C with bedding enrichment. At 8 weeks of age, mice were allocated to treatment groups to achieve matched mean body weight, and either continued on a normal chow diet (Harlan 8640) or were put on a HFHC (TD.09821) ± ETC-1002 targeted to achieve a 30 mg kg$^{-1}$ per day dosage for 12 weeks. Lesion histology and lipoprotein profile analyses were performed blinded, while the other study endpoints were not.

**Metabolic studies.** After 10 weeks of treatment, glucose tolerance tests were performed on 6 h fasted mice by intraperitoneal injection of glucose (1.2 g per kg body weight) and blood glucose measured by Aviva blood glucose monitor (Roche) from a small tail vein nick at the time points indicated. After 11 weeks of treatment, RER was assessed using a Columbus Laboratory Animal Monitoring System[64], and whole-body adiposity was measured by TD-NMR using a Bruker minispec. Fasting (12 h) and fed blood samples were collected by tail vein bleed for serum insulin measurements. Commercially available ELISA kits were used to measure plasma SAA (Tridelta Development Ltdl Kildare Ireland) and insulin (ALPCO Diagnostics, Salem, NH)

**Plasma lipids and lipoproteins.** At the end of study, fresh EDTA plasma (50 μl) from fasted (6 h) mice was separated by fast performance liquid chromatography using an AKTA purifier and a Superose 6 column. A constant flow rate of 0.4 ml min$^{-1}$ was used to collect 700 μl fractions. A 125 μl aliquot of each fraction was used to measure total cholesterol enzymatically in samples on a 96-well microtitre plate with 75 μl of two times concentrated reagents (triglyceride,

RocheDiagnostics, Laval, Quebec; cholesterol, WAKO Diagnostics, Richmond, VA; and standards, Randox, Crumlin, Co. Antrim, UK).

**Tissue histology.** Following perfusion fixation with 10% neutral buffered formalin[65,66], hearts (including the aortic roots) were cut transversely and embedded in paraffin. Serial sections, 4 μm thick, were cut starting from the aortic root origin and collected for measurement of lesion size (haematoxylin and eosin staining) and immunohistochemical analyses[67]. Lesions were traced manually and measured using computer-assisted image analysis equipment (Olympus BX41 microscope, Olympus DP70 charge-coupled device camera and ImagePro Plus software), and mean lesion size calculated from the first section of each animal. LDLR immunohistochemistry was performed on two sections from optimal cutting temperature (OCT) compound frozen liver samples from two mice per treatment group. Sections were incubated with diluted (1:20) goat anti-LDLR primary antibody (R&D Systems), followed by diluted (1:200) Alexa 488 donkey anti-goat secondary antibody. Blocking was performed in 5% normal donkey serum. Images were captured at × 20 magnification at identical exposure times.

**Tissue lipids.** Lipids were extracted from ~50 mg of frozen liver directly homogenized in 1.0 ml chloroform/methanol (2:1) or whole aortae with connective tissue removed, essentially described by Folch et al.[68] Samples were dried then solubilized in isopropanol and analysed for triglyceride (WAKO, Diagnostics, Burlington, Canada 11877771 216) or total cholesterol (WAKO, Cholesterol E, #439–17501) as per the manufacturer's instructions.

**Statistical analysis.** All values are reported as mean ± s.e.m. Data were analysed using Student's $t$-tests or one-way ANOVA or repeated measures ANOVA with Bonferroni *post hoc* tests where appropriate. Differences were considered significant when $P < 0.05$. Sample sizes were selected based on experience from our previous publications. Data were excluded only in the case where a technical error occurred in sample preparation or analysis.

**Data availability.** All relevant data are available from the authors.

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

## Acknowledgements

These studies were supported by grants from the Canadian Institutes of Health Research (G.R.S. and R.C.A.), Heart and Stroke Foundation of Canada (R.C.A.), and Esperion

Therapeutics (G.R.S.). E.A.D. was a recipient of an Ontario Graduate Scholarship and Queen Elizabeth II Graduate Scholarship in Science and Technology. B.K.S. is a recipient of a CIHR post-doctoral fellowship and McMaster University DeGroote Fellowship. R.C.A. is a Career Investigator of the Heart and Stroke Foundation of Ontario and holds the Amgen Canada Research Chair in the Division of Nephrology at St Joseph's Healthcare and McMaster University. G.R.S. is a Canada Research Chair in Metabolism and Obesity and the J. Bruce Duncan Endowed Chair in Metabolic Diseases at McMaster University. We thank Dr Murray W. Huff,  Dawn E. Telford and Cynthia G. Sawyez for their assistance in completing the fast performance liquid chromatography plasma lipoprotein analyses.

## Author contributions

S.L.P., R.S.N., R.C.A., S.F., P.H.E.G., N.D.L. and G.R.S. designed the experiments; S.L.P., E.A.D., R.J.F., S.L., C.M.B. and B.K.S. performed experiments and testing; S.L.P., E.A.D., R.J.F., S.L., C.M.B. and B.K.S. provided technical expertise and performed data analyses; S.L.P. and G.R.S. wrote the manuscript. All authors edited the manuscript and provided comments.

## Additional information

**Competing financial interests:** S.L.P., R.S.N., C.M.B., S.F., P.H.E.G., G.R.S. and N.D.L. received compensation from Esperion Therapeutics, Inc. The remaining authors declare no competing financial interests.

