## [Peer Review File · Nature Communications]

Reviewers' comments:

Reviewer #1 (Remarks to the Author):

Bempedoic acid (ETC-1002) is an ATP-citrate lyase (ACL) inhibitor that in clinical studies has been shown to decrease plasma LDL cholesterol apparently via decrease of cholesterol synthesis and upregulation of LDL receptor. ETC-1002 is actually a prodrug while the HSCoA ester is the active inhibitor. In this manuscript, the investigators demonstrate that the long chain acylcoenzyme A synthetase (ACSVL1) is the isoform responsible for CoA activation of ETC-1002 and that ACSVL1 is required for the activity of ETC-1002. Since ACSVL1 is not present in skeletal muscle, inhibition of isoprenoid synthesis in muscle would not be expected, potentially leading to less risk of myalgia symptoms and myopathy than is seen with statins. In addition, the authors demonstrate the LDL-lowering effect of ETC-1002 in a mouse model deficient in AMPK suggesting that LDL-lowering is independent of AMPK activation.

Specific comments:

1. Statins, but not other LDL-lowering therapies, are associated with muscle symptoms. Neither ezetimibe nor PCSK9 inhibitors have been shown to have a significantly higher rate of myotoxicity when compared with placebo. Thus, in abstract the last line myotoxicity commonly associated with "other lipid lowering therapies" should be changed to with "statins."
2. The introduction unnecessarily summarizes all of the results. It would be better to focus on the answered questions that these experiments address and wait for the results section to present the actual results.
3. It should be made clear how the cell-free enzymatic assays extend previous work published in Ref 26.
4. What gives ETC-1002-CoA its specificity for ACL inhibition? The bempedoic acid portion does not show similarity to citrate. Do the other CoA esters synthesized by ACSVL1 (ex palmitoyl or EPA) also inhibit ACL inhibition? The structure of bempedoic acid and the biochemical reaction of the CoA form inhibiting ACL should be shown in a figure.
5. The apoE KO mouse is an unusual model to explore LDL-lowering effects as it is primarily remnant particles, not LDL, that are increased in this model. How was LDL-C measured in the apoE KO experiments? FPLC was apparently performed and representative FPLC tracings on ETC-1002 or vehicle should be provided.
6. The hepatic LDLR was upregulated in the apoE KO mouse experiments as were other SREBP target genes upregulated consistent with a reduction in cholesterol synthesis? Is ETC-1002 effective in lowering LDL-C in LDLR KO mice consistent with a mechanism that is dependent on LDLR upregulation?
7. Is it possible that ACSVL1 is upregulated in ApoE^{-/-} mice because of reduction in normal products of this enzyme due to bempedoic acid?
8. Lesions in the DKO treated with ETC-1002 were not significantly lower. Why is the effect of ETC-1002 in the DKO attenuated compared with the apoE KO alone?
9. Genetic suppression of ACL expression in McArdle cells resulted in upregulation of LDLR expression and activity. The authors should go on to show that ETC-1002 has no further effect on cholesterol synthesis, LDLR expression or activity in the setting of ACL knockdown in order to formally prove that these effects of the compound are dependent on ACL activity. What is the effect of AMPK activation in this cell model on LDLR expression?
10. What happens to ETC-1002-CoA? It cannot be beta-oxidized due to presence of alpha methyl groups. Is it converted to TG or CE?

Reviewer #2 (Remarks to the Author):

This manuscript for the first time elucidates the mechanism by which bempedoic acid decreases LDL-c. This finding is important because it links the MOA with LDL receptor up regulation which has been consistently associated with CV event reduction (i.e. statins and ezetimibe and a point that needs to be emphasized in the discussion). The manuscript includes a number of preclinical experiments that methodically evaluated the MOA of this novel therapy for dyslipidemia. The writing is clear but the manuscript could benefit from a figure outlining the hepatic cholesterol pathway and the location of the bempedoic effect in the liver and muscle (and differentiate the tissues). A potential mechanistic difference between statins and this novel compound in the muscle is of high interest because one of the key potential clinical benefits of this therapy is in the treatment of dyslipidemia in patients with statin intolerance.

Reviewer #3 (Remarks to the Author):

A. Summary of the key results

Pinkosky and colleagues report that Bempedoic acid (ETC-1002) which is a novel chemical entity was previously shown to modulate both AMP-activated protein kinase (AMPK) activity and ATP-citrate lyase (ACL) in rodents. ETC-1002 is intended to lower LDL cholesterol in hypercholesterolemic patients. The authors have attempted to elucidate

- i) the mechanism by which Bempedoic acid lowers LDL cholesterol,
- ii) its relevance in humans, and
- iii) whether it reduces disease progression in models of atherosclerosis.

In the present report Pinkosky and colleagues demonstrate that Bempedoic acid is a prodrug that requires activation by liver very long-chain acyl-CoA synthetase 1 (ACSVL1) to directly modulate both molecular targets, and that inhibition of ACL leading to LDL receptor upregulation, decreased LDL cholesterol and the attenuation of atherosclerosis independent of AMPK.

In addition to establish a mechanistic basis for potentially avoiding the myotoxicity commonly associated with other lipid lowering therapies, the authors have demonstrated the absence of ACSVL1 expression in human skeletal muscle.

B.Originality and interest: if not novel, please give references

This is an original work.

C.Data & methodology: validity of approach, quality of data, quality of presentation

Satisfactory

D.Appropriate use of statistics and treatment of uncertainties

Sufficient

E.Conclusions: robustness, validity, reliability

Acceptable

F.Suggested improvements: experiments, data for possible revision

Demonstration of lack of undesirable adverse effects

G.References: appropriate credit to previous work?

Fair enough

H.Clarity and context: lucidity of abstract/summary, appropriateness of abstract, introduction and conclusions

Satisfactory

Reviewer #4 (Remarks to the Author):

In the present study the authors show that ETC-1002 lowers LDL-C independent of AMPK activity by using a mouse model system in which apoE and AMPK beta1 are knockout. The Ms is highly original and timely since it address the need for alternatives to standard statin therapy. The data are convincing and show further proof that the mechanism of action of this small molecule is restricted to the liver which again elevates the potential of this drug for use in individuals who suffer with muscle pain. The discussion is somewhat short but for a communication seems appropriate.

Reviewer 1:

Bempedoic acid (ETC-1002) is an ATP-citrate lyase (ACL) inhibitor that in clinical studies has been shown to decrease plasma LDL cholesterol apparently via decrease of cholesterol synthesis and upregulation of LDL receptor. ETC-1002 is actually a prodrug while the HSCoA ester is the active inhibitor. In this manuscript, the investigators demonstrate that the long chain acylcoenzyme A synthetase (ACSVL1) is the isoform responsible for CoA activation of ETC-1002 and that ACSVL1 is required for the activity of ETC-1002. Since ACSVL1 is not present in skeletal muscle, inhibition of isoprenoid synthesis in muscle would not be expected, potentially leading to less risk of myalgia symptoms and myopathy than is seen with statins. In addition, the authors demonstrate the LDL-lowering effect of ETC-1002 in a mouse model deficient in AMPK suggesting that LDL-lowering is independent of AMPK activation.

Specific comments:

1. Statins, but not other LDL-lowering therapies, are associated with muscle symptoms. Neither ezetimibe nor PCSK9 inhibitors have been shown to have a significantly higher rate of myotoxicity when compared with placebo. Thus, in abstract the last line myotoxicity commonly associated with "other lipid lowering therapies" should be changed to with "statins."

We agree with the reviewer. We have changed "...commonly associated with other lipid lowering therapies" to "...commonly associated with statin therapy." Page 2, lines 15 and 16.

2. The introduction unnecessarily summarizes all of the results. It would be better to focus on the answered questions that these experiments address and wait for the results section to present the actual results.

We have removed the last paragraph of the introduction that provided a detailed summary of the study results. We have also edited the final paragraph in the introduction to state the questions addressed by this study and high-level conclusions. Please see page 5, lines 6 through 10.

3. It should be made clear how the cell-free enzymatic assays extend previous work published in Ref 26.

We agree with the reviewer and have added the following statements to better characterize the importance of the findings from the cell-free enzyme analyses:

These findings in cell-free enzyme assays 1.) Establish that ETC-1002-CoA is a competitive ACL inhibitor with respect to free CoA, and 2.) Demonstrate that ETC-1002-CoA is the active form that directly interacts with AMPK, and not the free acid as previously proposed (26), and 3.) show that ETC-1002 can only inhibit ACL and activate AMPK β 1 in tissues that are capable of catalyzing the CoA activation of ETC-1002 depicted in (Figure 1G). Page 6, lines 7 through 11.

4. What gives ETC-1002-CoA its specificity for ACL inhibition? The bempedoic acid portion does not show

similarity to citrate. Do the other CoA esters synthesized by ASCVL1(ex palmitoyl or EPA) also inhibit ACL inhibition? The structure of bempedoic acid and the biochemical reaction of the CoA form inhibiting ACL should be shown in a figure.

Early work suggests that CoA thioesters derivatives of natural long-chain fatty acids (LCFA) inhibit ACL (Boulton and Ratledge J Gen Microb. 128, 1983) which would be consistent with a physiological product feedback mechanism that has evolved to prevent the accumulation of lipid products associated with the pathogenesis of metabolic disease. However, natural LCFA-CoA esters are rapidly metabolized or incorporated into CEs and TGs for storage, and unlikely to promote sustained inhibitory effects. Although ETC-1002 and natural LCFA share some common structural features that make them efficient ACS substrates, ETC-1002 seems to be resistant to β -oxidation and incorporation into other lipid species (see response to question 10) resulting in sustained inhibition. We have added the structure of ETC-1002 and the biochemical reaction for its ACS-dependent CoA activation to Figure 1G.

5. The apoE KO mouse is an unusual model to explore LDL-lowering effects as it is primarily remnant particles, not LDL, that are increased in this model. How was LDL-C measured in the apoE KO experiments? FPLC was apparently performed and representative FPLC tracings on ETC-1002 or vehicle should be provided.

We agree with the reviewer's comments regarding the ApoE model. However, given the evidence from our data suggesting that there was a link between inhibition of cholesterol synthesis by ETC-1002 and LDLR upregulation in primary human liver cells, we were interested in translating these findings to an LDLR competent in vivo model. Despite the fact that ApoE deficient mice carry the majority of their plasma cholesterol in the VLDL fraction, HFHC-feeding also elevated LDL-C, an effect that was markedly attenuated by ETC-1002 treatment and consistent with our cell-based assays (Figure 3E). To address the reviewer's request for the inclusion of a representative HPLC tracing, we have included an FPLC trace from each treatment group (Figure 3C).

6. The hepatic LDLR was upregulated in the apoE KO mouse experiments as were other SREBP target genes upregulated consistent with a reduction in cholesterol synthesis? Is ETC-1002 effective in lowering LDL-C in LDLR KO mice consistent with a mechanism that is dependent on LDLR upregulation?

The efficacy of ETC-1002 in LDLr KO mice is a very interesting question. While we agree that the findings in the LDLr KO mouse could be informative, caution must be taken when deriving mechanistic insights for the hypolipidemic effects of cholesterol synthesis inhibitors in this model. For example, Bisgaier et.

al. (J Lipid Research, 38, 1997) showed that atorvastatin significantly lowered LDL-C in LDLr KO mice, despite the lack of LDL receptors. Therefore, we believe that studying the effects of ETC-1002 in this model to specifically address LDLR dependence would yield ambiguous results.

7. Is it possible that ACSVL1 is upregulated in ApoE^{-/-} mice because of reduction in normal products of this enzyme due to bempedoic acid?

The reviewer raises a good point. Given the reductions in liver triglycerides and total cholesterol observed with ETC-1002 treatment, it is possible that a reduction in other lipid species, including ACSVL1 substrates, may contribute to its own upregulation. Although the transcriptional regulation of ACSVL1 is not well understood, it has been shown to be at least partially physiologically regulated by fasting and feeding. We have revised our statement to:

“Similar to srebf2, slc27a2 (ACSVL1) expression was increased by ETC-1002 treatment and in DKO mice compared to ApoE^{-/-} mice (Figure 3N), suggesting that slc27a2 may also be subject to transcriptional regulation by SREBP2. Although some ACS isoforms have been shown to be regulated by SREBP2(54), other effects on lipid metabolism resulting from loss of AMPK β1 or ETC-1002 treatment cannot be ruled out.”

Please see page 12, lines 14 through 18.

8. Lesions in the DKO treated with ETC-1002 were not significantly lower. Why is the effect of ETC-1002 in the DKO attenuated compared with the apoE KO alone?

Although LDL-C is elevated in the ApoE^{-/-} mouse, as the reviewer has previously stated, this model is primarily driven by the elevation of cholesterol in remnant particles. Therefore, the specific reduction of LDL-C by ETC-1002 leads to an expected modest reduction in total plasma cholesterol and a proportionate modest reduction in lesion size. The measure of aortic cholesterol provides a clearer picture of the effect throughout the whole aorta and clearly demonstrates a more robust reduction that is consistent with AMPK independent LDLr up-regulation and LDL-C lowering.

9. Genetic suppression of ACL expression in McArdle cells resulted in upregulation of LDLR expression and activity. The authors should go on to show that ETC-1002 has no further effect on cholesterol synthesis, LDLR expression or activity in the setting of ACL knockdown in order to formally prove that these effects of the compound are dependent on ACL activity. What is the effect of AMPK activation in this cell model on LDLR expression?

The reviewer asks a logical question. While we considered performing this experiment, we realized that this experiment presents several practical challenges and was not likely to address the specificity of ACL in mediating the effects of ETC-1002 on LDLR. Most importantly, since we cannot achieve 100% knockdown of ACL expression, treatment with ETC-1002 will most likely lead to additional suppression of cholesterol synthesis due to the inhibition of residual ACL activity. Furthermore, even if we were able to knock down ACL completely (and the hepatocytes survived which is unlikely given the lethality of

ACLY null mice), there would essentially be no flux through the lipid synthesis pathway, which would result in the saturation of LDLR expression and no assay window available to measure additional effects of any cholesterol synthesis inhibitor on LDLr. Given that in either case, this experiment would not be capable of discriminating between the effects of atorvastatin and ETC-1002, we believe that it is unlikely to extend the evidence supporting ACL as the molecular target of ETC-1002.

Given the above limitations we do agree with the reviewer that it is important to understand whether AMPK activation increases LDLR activity. We have directly addressed this by showing that treatment with A-769622, the β 1-selective AMPK activator, did not result in an increase in LDLR activity in human hepatocytes. Please see page 14 line 21 through page 15, line 5 and figure 5G. These findings support our observations in mice where we did not detect a difference in *ldlr* expression between apoE^{-/-} and DKO mice.

10. What happens to ETC-1002-CoA? It cannot be beta-oxidized due to presence of alpha methyl groups. Is it converted to TG or CE?

The conjugation of xenobiotic carboxylic acids to CoA esters is a common pathway utilized to prepare them for secretion into the bile. Once ETC-1002 is activated to ETC-1002-CoA in liver, we know that it is not incorporated into CE or TGs. A representative TLC image comparing lipid extracts from primary rodent hepatocytes exposed to [¹⁴C]-ETC-1002 or [¹⁴C]-palmitate has been included below to highlight this point. Since ETC-1002 cannot be beta-oxidized or stored, it must be eliminated. The primary ETC-1002 metabolite observed that requires CoA activation is a taurine ETC-1002 conjugate which is eliminated in the feces.

Reviewer #2 (Remarks to the Author):

This manuscript for the first time elucidates the mechanism by which bempedoic acid decreases LDL-c. This finding is important because it links the MOA with LDL receptor up regulation which has been consistently associated with CV event reduction (i.e. statins and ezetimibe and a point that needs to be emphasized in the discussion). The manuscript includes a number of preclinical experiments that methodically evaluated the MOA of this novel therapy for dyslipidemia. The writing is clear but the manuscript could benefit from a figure outlining the hepatic cholesterol pathway and the location of the bempedoic effect in the liver and muscle (and differentiate the tissues). A potential mechanistic difference between statins and this novel compound in the muscle is of high interest because one of the key potential clinical benefits of this therapy is in the treatment of dyslipidemia in patients with statin intolerance.

We thank the reviewer for their thoughtful review and suggestions. We agree that we have not adequately emphasized the established link between LDLR upregulation and CVD risk reduction. We have added the following statement to the discussion:

“As such, the management of LDL-C levels can also be achieved by other mechanisms such as inhibition of cholesterol absorption in the gut (i.e. ezetimibe), or preventing LDL-receptor degradation (i.e., PCSK9 inhibitors).(57-59) Importantly, each of these mechanisms primarily reduces LDL-C by upregulating the activity of the LDLR, a mechanism proven to reduce CV events.(60-62).”

Please see page 17, lines 17 through 21.

We have also added the following figure to outline and contrast the effect of ETC-1002, and statins, in liver and muscle. Please see page 18, lines 4 through 7, and figure 6.

Reviewer #3 (Remarks to the Author):

A. Summary of the key results

Pinkosky and colleagues report that Bempedoic acid (ETC-1002) which is a novel chemical entity was previously shown to modulate both AMP-activated protein kinase (AMPK) activity and ATP-citrate lyase (ACL) in rodents. ETC-1002 is intended to lower LDL cholesterol in hypercholesterolemic patients. The authors have attempted to elucidate

- i) the mechanism by which Bempedoic acid lowers LDL cholesterol,
- ii) its relevance in humans, and
- iii) whether it reduces disease progression in models of atherosclerosis.

In the present report Pinkosky and colleagues demonstrate that Bempedoic acid is a prodrug that requires activation by liver very long-chain acyl-CoA synthetase 1 (ACSVL1) to directly modulate both molecular targets, and that inhibition of ACL leading to LDL receptor upregulation, decreased LDL cholesterol and the attenuation of atherosclerosis independent of AMPK.

In addition to establish a mechanistic basis for potentially avoiding the myotoxicity commonly associated with other lipid lowering therapies, the authors have demonstrated the absence of ACSVL1 expression in human skeletal muscle.

B.Originality and interest: if not novel, please give references

This is an original work.

C.Data & methodology: validity of approach, quality of data, quality of presentation

Satisfactory

D.Appropriate use of statistics and treatment of uncertainties

Sufficient

E.Conclusions: robustness, validity, reliability

Acceptable

F.Suggested improvements: experiments, data for possible revision

Demonstration of lack of undesirable adverse effects

G.References: appropriate credit to previous work?

Fair enough

H.Clarity and context: lucidity of abstract/summary,

appropriateness of abstract, introduction and conclusions

Satisfactory

We thank the reviewer for their comments. We believe that the strongest evidence supporting a lack of undesirable adverse effects is found in the clinical safety data. Please see the below references which describe the efficacy and safety of ETC-1002 in approximately 1000 patients.

- 1 Ballantyne, C. M. et al. Efficacy and safety of a novel dual modulator of adenosine triphosphate-citrate lyase and adenosine monophosphate-activated protein kinase in patients with hypercholesterolemia: results of a multicenter, randomized, double-blind, placebo-controlled, parallel-group trial. *J Am Coll Cardiol* 62, 1154-1162, doi:10.1016/j.jacc.2013.05.050 (2013).
- 2 Ballantyne, C. M. et al. Effect of ETC-1002 on Serum Low-Density Lipoprotein Cholesterol in Hypercholesterolemic Patients Receiving Statin Therapy. *Am J Cardiol* 117, 1928-1933, doi:10.1016/j.amjcard.2016.03.043 (2016).
- 3 Gutierrez, M. J. et al. Efficacy and safety of ETC-1002, a novel investigational low-density lipoprotein-cholesterol-lowering therapy for the treatment of patients with hypercholesterolemia and type 2 diabetes mellitus. *Arterioscler Thromb Vasc Biol* 34, 676-683, doi:10.1161/atvbaha.113.302677 (2014).
- 4 Lemus, H. N. & Mendivil, C. O. Adenosine triphosphate citrate lyase: Emerging target in the treatment of dyslipidemia. *J Clin Lipidol* 9, 384-389, doi:10.1016/j.jacl.2015.01.002 (2015).

- 5 Thompson, P. D. et al. Treatment with ETC-1002 alone and in combination with ezetimibe lowers LDL cholesterol in hypercholesterolemic patients with or without statin intolerance. *J Clin Lipidol* 10, 556-567, doi:10.1016/j.jacl.2015.12.025 (2016).
- 6 Thompson, P. D. et al. Use of ETC-1002 to treat hypercholesterolemia in patients with statin intolerance. *J Clin Lipidol* 9, 295-304, doi:10.1016/j.jacl.2015.03.003 (2015).

Reviewer #4 (Remarks to the Author):

In the present study the authors show that ETC-1002 lowers LDL-C independent of AMPK activity by using a mouse model system in which apoE and AMPK beta1 are knockout. The Ms is highly original and timely since it address the need for alternatives to standard statin therapy. The data are convincing and show further proof that the mechanism of action of this small molecule is restricted to the liver which again elevates the potential of this drug for use in individuals who suffer with muscle pain. The discussion is somewhat short but for a communication seems appropriate.

We thank the reviewer for their positive comments stating that the manuscript is original, high quality and important.